# E-Cigarette Aerosols Promote Oral *S. aureus* Colonization by Delaying an Immune Response and Bacterial Clearing

**DOI:** 10.3390/cells11050773

**Published:** 2022-02-23

**Authors:** Alma R. Cátala-Valentín, Jasmine Almeda, Joshua N. Bernard, Alexander M. Cole, Amy L. Cole, Sean D. Moore, Claudia D. Andl

**Affiliations:** Burnett School of Biomedical Sciences, University of Central Florida College of Medicine, Orlando, FL 32816, USA; alma.catala@knights.ucf.edu (A.R.C.-V.); jasminealmeda8@knights.ucf.edu (J.A.); joshua.bernard@ucf.edu (J.N.B.); acole@ucf.edu (A.M.C.); amy.cole@ucf.edu (A.L.C.); sean.moore@ucf.edu (S.D.M.)

**Keywords:** inflammation, oral cavity, bacterial–epithelial interactions, *S. aureus*, COX2

## Abstract

E-cigarette (e-cig) vapor has been shown to play a pathological role in oral health and alter the oral microbiota, providing growth advantages for opportunistic pathogens. Enrichment of *Staphylococcus aureus*, a commensal resident in the oral cavity, correlates with the progression of periodontal disease, suggesting a role as an opportunistic pathogen. Environmental conditions, such as cigarette smoke, are known to increase *S. aureus* virulence, yet the role of *S. aureus* in periodontitis and oral preneoplasia is unknown. We exposed oral epithelial cells to e-cig aerosols and showed a dose-dependent cell viability reduction, regardless of nicotine content, in a possible attempt to repair DNA damage, as measured by pH2AX. *S. aureus* attachment to oral epithelial cells and bacterial biofilm formation were enhanced upon e-cig exposure, indicating an increased capacity for oral colonization. Mechanistically, e-cig aerosol exposure resulted in an immunosuppression, as determined by a reduction in IL8, IL6, and IL1β secretion by oral epithelial cells during co-culture with *S. aureus*. Consistent with this, e-cig vape reduced the oral epithelial cell clearance of *S. aureus*. Furthermore, we observed an increased expression of the inflammatory regulator COX2. This work suggests that e-cigs promote *S. aureus* colonization and modulate the oral inflammatory response, possibly promoting oral periodontitis and preneoplasia.

## 1. Introduction

E-cigarettes (e-cigs) have gained popularity during the last decade, especially among adolescents and young adults (ages 18–24) [1,2]. In 2020, the National Youth Tobacco Survey (NYTS) reported that 19.6% of high school students and 4.7% of middle school students are current e-cigarette users. Being advertised as less harmful than conventional cigarettes [3], the reported consequences of e-cig vaping on oral health are inconsistent [4,5]. While some epidemiological studies have shown that e-cig users and never users have similar risks of developing periodontal disease [6], other studies reported an increase in gum disease in current e-cig users compared with never users [7]. Oral disease symptoms are greater in cigarette smokers compared with e-cig users, but e-cig users still experience more oral discomfort, increased periodontal symptoms (e.g., probing depth, bone loss, plaque index, inflammation), and overall poor oral health compared with never users [4]. In contrast to conventional cigarette smoke, the levels of carcinogenic compounds found in e-cigs aerosols are lower, but these lower levels of toxicants are still not safe and can include the probable human carcinogens—formaldehyde and acetaldehyde [5,8]. Overall, e-cig aerosols are rarely reported to induce tissue damage in oral cells in vitro (e.g., apoptosis, cell alteration, keratinization), but it has been shown that e-cig aerosols induce an inflammatory response in buccal cultures [8] and significantly higher levels of proinflammatory cytokines [4].

E-liquids base compounds are propylene glycol and vegetable glycerin, with further additives, such as nicotine and flavorants [9]. When heated to high combustible temperatures by e-cigarette devices, e-liquids produce e-cigarette aerosol, containing various toxicants and carcinogens, including formaldehyde, acetaldehyde, and acrolein—as a result of the dehydration and fragmentation of glycerin [10]—and nitrosamines, produced during the nitrosation of nicotine [11]. The puff behavior or topography affects the coil temperature, and therefore affects the potential chemicals that can be formed during the burning of e-liquids, since longer puff durations can result in higher emission of toxicants and carcinogens [12]. While not a carcinogen in itself, nicotine exposure induces DNA damage, inflammation through NF-kB, proliferation, and supports stemness and progenitor cell expansion [13,14]. The inflammation, driven by NF-kB, increases the transcription of proinflammatory cytokines, creating a tumor microenvironment thought to play a key role in tumorigenesis [15]. Tobacco-derived nitrosamines, such as N-nitrosonornicotine and 4-(methylnitrosamino)1-(3-pyridyl)-1-butanone, have been shown to be present in the saliva of e-cig users [16]. These chemicals promote proliferation, survival, and activate various inflammatory signaling pathways, including NF-kB, Erk1/2, and STAT1 [11,17].

Cigarette smoke, as well as e-cig aerosols, are both known to alter the oral microbiota composition, leading to an enrichment in periodontal pathogens [18,19]. In addition, the propylene glycol present in vape liquid has antimicrobial properties: when tested for its bactericidal activity against 5 common oral bacteria, propylene glycol ranked after polyethylene glycol 1000 in its effectiveness, yet neither could kill *S**. aureus* [20]. A link between e-cig vaping and *S. aureus* colonization has been reported previously, as exposure of human airway epithelial cells to e-cig vape at the air liquid interface in vitro showed reduced antimicrobial activity in the host cells and the expression of inflammatory markers [21,22]. Among the proinflammatory cytokines that form part of the inflammatory response are IL8 (a neutrophil chemoattractant) and IL6, which induce the transition from a neutrophil inflammation to a monocyte-type inflammation, and also stimulates T-cells and B-cells as part of the adaptive immune response [23]. Additionally, the proinflammatory cytokines IL1β and TNFα lead to the activation of NF-kB, JNK, and MAPK signaling [23].

Furthermore, inflammation sensitizes tissues to homologous recombination, which, in constantly regenerating cells, leads to an increased risk of mutagenesis and malignant transformation [24]. Inflammation in the tumor microenvironment stimulates cell survival and proliferation, promoting cancer progression [24] through the release of reactive oxygen species and reactive nitrogen intermediates that cause DNA damage in the cells and enhance the inflammatory response even further [24].

Here, we focus on the effect of e-cig aerosols on *Staphylococcus aureus*, which is known to be enriched in the oral cavity of periodontitis patients [25,26,27]. We aimed to understand the role of e-cig aerosols in the modulation of the interaction between oral epithelial cells and *S. aureus*, and subsequently the role *S. aureus* plays in the development of inflammatory processes linked to periodontitis. We hypothesized that e-cig aerosols have the capacity to enhance the colonization of *S. aureus* and consequently exacerbate the host inflammatory signaling.

## 2. Materials and Methods

### 2.1. Cultivation of Cell Lines

Human oral epithelial cells (OKF6) [28] and human esophageal epithelial cells (STR, EPC-h-TERT) [29] were cultured in keratinocyte serum-free medium (Life Technologies Co 10724-011, Grand Island, NY, USA), supplemented with epithelial growth factor (1 ng/mL), bovine pituitary extract (0.05 mg/mL), and 1% penicillin/streptomycin (Gibco^TM^ Life Technologies Co 15140-122, Carlsbad, CA, USA). Cells were incubated at 37 °C with 5% CO_2_.

### 2.2. Bacterial Cultures

*Staphylococcus aureus* strains 512 (sequence type ST30) and 528-11 (sequence type ST8/t008), 547, S042, and S072 were used in this study [30]. Strains 512 (sequence type ST30) and 528-11 (sequence type ST8/t008) have been reported to be common isolates from the oral cavity and have been associated with active sites of periodontitis [31]. All strains were grown in tryptic soy broth (TSB; BD&Co 211825, Sparks, MD, USA) or KSFM supplemented with 1% FBS, EGF, and BPE, at 37 °C and 5% CO_2_. Bacteria were grown on trypticase soy blood agar plates (TSA; BBL^TM^ 221261, Sparks, MD, USA) and sub-cultured from an independent colony in TSB. Bacteria were snap-frozen in liquid nitrogen for 30 min during the exponential growth phase, and stored at −80 °C. To determine CFU/mL, bacteria were serially diluted in PBS, and spot plated (20 µL) on TSA blood agar plates.

### 2.3. Electronic Cigarette Aerosol

Initially, e-cigarette liquid (a 50/50 mix of vegetable glycerin to propylene glycol as base purchased from Vaporfi (Miami Lakes, FL, USA)) was vaporized in the custom-built chamber created using a Intex Quick-Fill AC Electric Air Pump connected to a Komax Biokips Extra Large Container, and a SMOK^®^ G-PRIV Baby e-cigarette. Nicotine-free e-cigarette vape and vape containing 3 mg/mL nicotine were aerosolized for a 10 s puff in the vape chamber and compared with air in the vape chamber and incubator controls. For the air control conditions, the pump was run for 10 s with no vapor. Following the 10 s puff, cells were incubated within the vape chamber for 5 min, and then returned to proper incubation conditions within an incubator. As indicated, most experiments were performed using commercially available e-liquid without flavoring purchased from a different vendor (Vapor Vapes, Sand City, CA, USA), without nicotine (nicotine free—NF), and with 3 mg/mL of nicotine (nicotine—Nic; concentration as reported by manufacturer), both with a mixture of 50% propylene glycol/50% vegetable glycerol (50/50 PG/VG). E-liquid was vaporized through heat generated by a fourth-generation e-cig (SMOK^®^ G-PRIV Baby Kit). KSFM media (10 mL in a 10 cm dish) were pretreated with e-cig aerosols before every experiment. Based on the known e-cig puff topography of established cigarette smokers [12], we pretreated the media with a session of 3, 6, or 15 puffs: 3 s puff length and 30 s intervals between puffs. Epithelial cells were exposed to e-cig aerosol pretreated media (6 puffs) for 30 min; *S. aureus* strains were exposed to e-cig aerosol pretreated media for 24 h. E-cig exposure was performed at 37 °C with 5% CO_2_.

### 2.4. Bacterial and Epithelial Cells Co-Culture

Bacteria and epithelial cells were co-cultured in KSFM supplemented with EGF (1 ng/mL) and BPE (0.05 mg/mL), and without antibiotics. The multiplicity of infection (MOI) utilized were between 1 (1:1 ratio) and 10^−2^ (1:100, bacteria:cell ratio).

### 2.5. Epithelial and Bacterial Cell Viability

#### 2.5.1. Cell Titer Blue

Epithelial cells (2.0 × 10^3^ cells/mL seeding density in a 96-well plate; 100 µL per well) were treated with e-cigarette aerosol pretreated media for 30 min or continuously. The media was pretreated with 3, 6, and 15 puffs or nicotine free and nicotine e-cig aerosol. Treated cells were incubated with cell titer-blue reagent provided in the cell viability assay (10 µL per well; Promega G8081, Madison, WI, USA) for 1.5 h. Fluorescence is recorded at 560_Ex_/590_Em_ using a Spectramax i3x Multi-Mode Detection system (Molecular Devices, San Jose, CA, USA) according to the manufacturer’s protocol. The manufacturer notes a linear relationship between cell number and fluorescence.

#### 2.5.2. Click-iT EdU

Human epithelial cells were seeded (50,000 cells/well) in chamber slides (Nalge Nunc International 154526, Rochester, NY, USA), and treated with e-cig vapor for 30 min. After e-cig treatment, cells were washed with PBS, and stained with EdU (10 µM) for 6 h, following the manufacturer’s protocol for the Click-iT EdU kit (Invitrogen C10269). Slides were incubated with Hoechst DNA stain at 4 °C overnight, and mounted with mounting medium (Southern Biotech 0100-01, Roskilde, Denmark). Alexa fluor 488-positive cells were enumerated and a ratio over total cells (Hoechst-positive) was calculated.

#### 2.5.3. BacTiter Glow

To observe the viability changes caused by e-cig vape during exponential phase and stationary phase, bacterial overnight cultures were diluted (1:10) into e-cig aerosol pretreated media (6 puffs), loaded into a 96-well plate (100 uL per well), and allowed to grow exponentially. Following the manufacturer protocol, viability was measured every hour with BacTiter-Glo Microbial Cell Viability Assay (100 uL per well; Promega G8232, Madison, WI, USA), which measures viability based on quantitation of the ATP present in the bacterial cells. The reagent was added to bacterial cells, and luminescence was measured using the Spectramax i3x Multi-Mode Detection system (Molecular Devices, San Jose, CA, USA). 

### 2.6. Attachment and Clearing Assays

Changes in bacterial attachment, which is the first step during bacterial invasion, was measured following a modified protocol previously described [32]. OKF6 cells were seeded at 2.5 × 10^5^ cells/mL in 24-well plates and incubated with e-cig pretreated *S. aureus* for 3 h to allow attachment to epithelial cells. The supernatant was collected, remaining attached cells were washed twice with 1×PBS to eliminate any non-attached bacteria, and 1×PBS was also collected. Then, the epithelial cells with possible internalized bacteria and attached bacteria were permeabilized by incubation in 1% Triton X-100 (100 µL for 10 min), the Triton X-100 was then diluted by adding 900 uL of sterile 1×PBS. All the collected bacteria were centrifuged (13,000× *g* for 5 min). Bacteria were then resuspended in 1 mL of 1×PBS and incubated with BacLight Green Bacteria stain (100 µL per well; Invitrogen B35000, Eugene, OR, USA) (1:1000 dilution) for 15 min. Bacteria luminescence values were normalized to luminescence of uninfected permeabilized epithelial cells. After incubation, bacteria were loaded into a 96-well plate, and luminescence was measured with the Spectramax i3x Multi-Mode Detection system (Molecular Devices, San Jose, CA, USA).

### 2.7. Biofilm Formation

Biofilm formation was measured following a modified protocol previously described [33]. TSB was pretreated with e-cigarette aerosols (6 puffs), overnight cultures were diluted 1:100 into pretreated media and incubated for 24 h in a 96-well plate (100 µL per well). The wells were washed with distilled water to remove planktonic bacteria and incubated at 37 °C to air dry for 20 min. The biofilm was incubated in 125 µL of safranin (Millipore Sigma 65092B-95, Darmstadt, Germany) at room temperature for 20 min. Safranin was discarded, and the wells were washed with distilled water, and air dried at 37 °C. The biofilm was fixed and dissociated by resuspending in a mix of ethanol and acetone (80:20) and absorbance measured at OD 490 nm using the Spectramax i3x Multi-Mode Detection system (Molecular Devices, San Jose, CA, USA). We set the OD for the control to 1 to allow for a direct comparison of the effect of treatments.

### 2.8. ELISA

Conditioned media were collected and stored at −80 °C immediately after collection. The levels of inflammatory markers, including IL8 (DY208), IL1β (DY201), TNFα (DY210), and IL6 (DY206), were measured by ELISA (R&D Systems). Conditioned media from co-cultures were centrifuged to pellet the remaining bacteria. ELISA components were reconstituted following the manufacturer’s certificate of analysis and a BD Falcon Microtest 96-well ELISA plate (R&D Systems DY990, Minneapolis, MN, USA) was coated with the corresponding capture antibody (100 µL per well) and incubated overnight at room temperature according to the manufacturer’s guidelines. Standards and samples were loaded into 96-well plate (100 µL per well) and incubated at room temperature for 2 h before detection with horse radish peroxidase-conjugated antibody (1:40; 100 µL per well) and substrates A:B (R&D Systems DY999, Minneapolis, MN, USA). Reaction was stopped with 2N H_2_SO_4_. Using the Spectramax i3x Multi-Mode Detection system (Molecular Devices, San Jose, CA, USA), and the absorbance was measured at OD 450 nm, with a wavelength correction set at 540 nm. Standard curves were plotted, and sample values were interpolated using a four-parameter logistic curve fit (second order polynomial quadratic curve) on GraphPad Prism (Prism 8).

### 2.9. qRT-PCR

Total RNA was collected from 6-well plates with QIAzol Lysis Reagent (500 µL; Qiagen 1023537, Germantown, MD, USA), vortexed, and stored at −80 °C. Phenol/Chloroform RNA extraction was performed (100 µL of chloroform per sample), followed by RNA cleanup/purification using a miRNeasy Mini Kit (Qiagen 217004, Hilden, Germany), following manufacturer’s protocol. Genomic DNA contamination was eliminated using the gDNA wipeout provided by the QuantiTect Reverse Transcription Kit (Qiagen 2052113, Hilden, Germany), and 500 ng of RNA was used to prepare complementary DNA (cDNA). cDNA was diluted 1:5 (5 ng/µL final concentration), and stored at −80 °C.

qRT-PCR was performed using QuantiFast SYBR Green (Qiagen 204056, Hilden, Germany), following the manufacturer’s protocol. Target genes primer sequences were designed (10 mM; Integrated DNA Technologies, Coralville, IA, USA; Table 1), and the genes Rpl13a (Qiagen QT01850576) and Eif3d (Qiagen QT00010829) were used as housekeeping genes. The comparative Ct methods was utilized to calculate the fold change in mRNA expression.

### 2.10. Immunofluorescence Staining


Human epithelial cells were seeded (50,000 cells/well) in chamber slides (Nalge Nunc International 154526, Rochester, NY, USA), and treated with e-cig aerosolized media for 30 min. After e-cig treatment, cells were washed with PBS, fixed with methanol: acetone (1:1) for 10 min at room temperature. After fixation, cells were permeabilized with 0.5% Triton X-100 for 20 min and blocked with 1% BSA for 1 h. Slides were incubated with primary antibody overnight at 4 °C (pERK1/2 1:250; NF-KB 1:400; p-H2A.X 1:400, all antibody from Cell Signaling, Danvers, MA, USA), washed with PBS, and incubated with corresponding secondary antibody (Alexa Fluor 488 goat anti-mouse, Invitrogen by ThermoFisher A11001, Eugene, OR, USA; Goat anti-rabbit IgG DyLight 488 Conjugated, Invitrogen 35552, Rockford, IL, USA) overnight at 4 °C. Slides were mounted with DAPI (Southern Biotech 0100-20, Roskilde, Denmark). 

### 2.11. Statistical Analyses

Each experiment was performed in three biological replicates with technical triplicates, *n* = 3. The differences among the groups were assessed by one-way ANOVA, with Dunnett’s correction compared with the corresponding control. Student’s *t*-test was used to compare the means of two groups. *p* ≤ 0.05 was considered significant. The values are expressed as mean ± standard error of the mean (SEM).

## 3. Results

### 3.1. E-Cig Vape Activates Proinflammatory pERK½ and NF-KB Signaling and Induces DNA Damage

To evaluate the effect of e-cig vape on epithelial cell lines from the oral cavity and upper aerodigestive tract, OKF6 and STR cells were initially placed in a custom vape chamber and exposed to e-cig liquid vaporized in the chamber. Nicotine-free e-cig vape and vape containing 3 mg/mL nicotine in the vape chamber were compared with air in the vape chamber and incubator controls. We observed a significant suppression in cell proliferation when cells were exposed to e-cig vape containing nicotine as assessed by EdU incorporation (Figure 1A,B). We further confirmed reports [34] that e-cig vape induces the expression of the inflammatory mediator cyclooxygenase-2 (COX2), more so when the e-cig vape contained nicotine (Figure 1C). Activation of pERK1/2 signaling by e-cig vape has been described previously [35], which we confirmed by Western blot (Figure 1C). Additionally, the nuclear translocation of pERK1/2 and NF-kB was demonstrated by immunofluorescence staining (Figure 1D). Together, these data verified that e-cig vape, regardless of nicotine content, induced proinflammatory signaling and reduced cell proliferation upon acute e-cig vape exposure.

To assess whether the reduced cell proliferation was associated with the attempt of cells to repair DNA damage upon e-cig vape exposure, we used an antibody against the DNA damage marker pH2AX, which allows the identification of double-stranded DNA breaks [36,37]. Quantification of pH2AX-positive cells indicated that e-cig vape exposure induced DNA damage (Figure 2).

### 3.2. Acute Exposure to E-Cig Aerosol Decreases Epithelial Cell Viability in a Dose-Dependent Manner

To allow for long-term exposure mimicking habitual vaping, media were pre-exposed to e-cig aerosols with 3, 6, or 15 puffs, each puff was 3 s long with 30 s intervals between puffs. Epithelial cells were then exposed to e-cig aerosol pretreated media for 30 min, followed by a 48 h recovery period and cell viability was measured every 24 h by cell titer blue. E-cig aerosols reduced cell viability in a dose-dependent manner, and a complete cytostatic effect was induced by the 15-puff treatment. After the 6-puff treatment, the cells recovered within 24 h. In addition, the cells remained overall unaffected by the 3-puff treatment compared to the control (Figure 3A,B). We also observed that a continuous exposure for 48 h to the e-cig pretreated media had a cytotoxic effect on the epithelial cells (Figure 3C,D). The findings of reduced viability and proliferation not only verified that the effect is dose-dependent, but also that the exposure of aerosol pretreated media had the same effect as direct exposure to the aerosolized vape in the chamber.

### 3.3. E-Cig Aerosols Do Not Alter S. aureus Growth but Promote Biofilm Formation and Attachment to Oral Cells Aiding in Oral Colonization

For our study, we selected the sequence types ST30 (512) and ST8/t008 (528-11) strains which have been reported to be prevalent isolates from the oral cavity, with 528-11 being less abundant than 512 and associated with active sites of periodontitis [31]. When exposed to e-cig vape, *S. aureus* upregulated genes responsible for virulence and resistance to antimicrobial peptides [21]. We showed that chronic exposure to e-cig aerosol pretreated media does not reduce the growth rate or viability of two different strains of *S. aureus*, 512 and 528-11, isolated from healthy nasal carriers (Figure 4A,B). While *S. aureus* remained unaffected by e-cig exposure, the abundance of other oral bacteria is known to decrease after e-cig exposure, causing dysbiosis [19]. This alteration in the homeostasis of the oral microbiome could confer *S. aureus* an advantage over other bacteria by leaving a niche for colonization. We therefore aimed to understand the effect of e-cig aerosol on *S. aureus* biofilm formation and colonization capacity. In the oral cavity, bacteria colonize the teeth by forming biofilms known as dental plaque [38]. We explored the effect of e-cig aerosols on *S. aureus* biofilm formation and observed that e-cig aerosols with and without nicotine significantly enhanced *S. aureus* (512 and 528-11) biofilm formation (Figure 4C,D).

Attachment is considered the initial step in the colonization process [26]. Therefore, next, we pre-exposed *S. aureus* to e-cig aerosol pretreated media and co-cultured pre-exposed bacteria with oral epithelial cells (MOI 1) to measure changes in *S. aureus* attachment to oral epithelial cells. We determined that e-cig pretreated *S. aureus* 512 had a higher attachment capacity to epithelial cells compared with unexposed *S. aureus* controls (Figure 5A) but not 528, which showed reduced attachment (Figure 5B). When assessing the attachment capacities of multiple *S. aureus* strains and sequence types including strains isolated from the nasal flares from smokers, we found that different *S. aureus* strains have varying attachment capabilities (Appendix A Appendix A).

To study the effect of e-cig exposure on the capacity of oral epithelial cells to clear *S. aureus* infection, OKF6 were pre-exposed to e-cig pretreated media (30 min), followed by inoculation with pre-exposed *S. aureus* strains (MOI 1). We observed that e-cig pretreated OKF6 cells showed a lower capacity to clear both *S. aureus* strains compared with the untreated co-culture after 3 h (Figure 5C,D). These results suggest that e-cig exposure allows bacteria to survive in the host, conferring *S. aureus* an advantage to colonize the oral cavity after e-cig exposure.

### 3.4. E-Cig Aerosol Decreased the Release of Proinflammatory Cytokines IL8, IL6, and IL1β

The finding that e-cig aerosols enhanced *S. aureus* oral colonization led us to question how *S. aureus* may evade an immune response. Epithelial cells can release β-defensins and other antimicrobials as an initial defense, along with other proinflammatory cytokines, helping recruit immune cells to the area of infection [23]. To determine the role of e-cig exposure on the release of these cytokines upon *S. aureus* infection, we pre-exposed OKF6 cells to e-cig pretreated media (30 min), followed by inoculation with healthy *S. aureus* strains at MOI 10^−3^ and 10^−2^. OKF6 co-cultured with *S. aureus* 512 at MOI 10^−3^ showed an increased release of IL8 and IL1β at later time points (Appendix A). After inoculation with *S. aureus* strains at MOI 10^−2^, conditioned media was collected for the analysis of cytokine secretion by ELISA (Figure 6). The results of the cytokine release for IL8 and IL1β were confirmed by qRT-PCR (Appendix A Appendix A). E-cig pre-exposed media, without bacteria, did not induce the secretion of any cytokines and COX2 (Figure 6A,D,G,J,M). Exposure to *S. aureus* 512 strain induced a cytokine release of IL1β, IL8, IL6, and TNFα (Figure 6B,E,H,K) compared with uninfected control and e-cig pre-exposed media only. E-cig pre-exposure caused a trend in the decrease in IL1β, IL8, and IL6 release compared with the control co-culture with *S. aureus* 512 strain (Figure 6E,H). Secretion of TNFα and COX2 expression were enhanced under all conditions containing *S. aureus*. The 528-11 strain (Figure 6 C,F,G,I,L) showed a significant induction of the inflammatory cytokine IL1β with *S. aureus* under nicotine-containing conditions, but only a minimal increase in IL8, IL6, and TNFα. We speculate that the observed discrepancy in results between 512 and 528 is due to differences in bacterial–cell interactions resulting in less secretion of cytokines after co-culture with 528-11. Strain 528-11 is known to be a resident of the oral cavity and therefore may not induce as strong of an inflammatory response [21]. In addition, Cole et al. also determined previously that the SpA content of 528-11 was low compared with most nasal *S. aureus* strains [39]. Overall, COX2 was increased significantly in e-cig aerosol conditions combined with both strains 512 and 528-11. COX2 is considered a central player in the mediation and induction of an inflammatory response and promotes apoptotic resistance, proliferation, angiogenesis, and invasion [40]. The inflammatory cytokines TNFα and IL1β are both known to induce COX2 expression, and we observed the same correlation in our experiment, suggesting the activation of COX2 by TNFα and IL1β.

### 3.5. DNA Damage Is Further Enhanced in the Presence of E-Cig Aerosols and S. aureus

During chronic exposure to e-cig aerosols, a COX2-induced increase in proliferation and decrease in apoptosis associated with DNA damage could lead to the accumulation of mutations and subsequently cancer initiation [41,42]. E-cig vape is known to induce DNA damage [14,43]. While we showed that the initial (acute) exposure to e-cig vape induces DNA damage (Figure 2), here, we analyzed DNA damage in response to e-cig aerosols and *S. aureus* by immunofluorescence staining of the DNA damage marker, pH2AX (Figure 7A). We exposed OKF6 cells to e-cig pretreated media, followed by co-culture with *S. aureus* strains (MOI1). We observed that e-cig aerosol with nicotine significantly induced DNA damage compared with control (Figure 7B). Furthermore, both *S. aureus* strains also induced DNA damage compared with control and regardless of e-cig aerosol exposure (Figure 7C). These results confirm that *S. aureus* and e-cig exposure both induce DNA damage in epithelial cells, which could play an important role in DNA damage accumulation during chronic e-cig usage.

## 4. Discussion

We show that e-cig aerosol exposure enhanced *S. aureus* biofilm formation and its attachment to oral epithelial cells. E-cig aerosols also affected the capacity of the host to clear *S. aureus* infection, consistent with a decrease in the secretion of proinflammatory cytokines. Finally, we show how e-cig aerosols can exacerbate the inflammatory signaling after the initial exposure, as observed by an increase in COX2 expression. Understanding the role of e-cig vape on the behavior of opportunistic pathogens and their interactions with the oral epithelium will shed light on the potential risks of e-cig vaping in the development of periodontitis, oral premalignant lesions, and subsequently oral cancer initiation.

Cole and colleagues previously demonstrated that cigarette smoke impairs the clearance of *S. aureus* in individuals using traditional cigarettes. Higher *S. aureus* nasal loads were found in cigarette smokers compared with healthy non-smokers and were reduced upon participation in a smoking cessation program [44]. They also observed that successful clearance was associated with an increase in inflammatory markers, including G-CSF and IL1β, which are part of the mucosal defense response [44]. Our current in vitro model using e-cig aerosols highlights the role IL1β signaling plays in the response to *S. aureus* carriage (Figure 6). Here, we established co-cultures of normal oral epithelial cells, OKF6, with two strains of *S. aureus* isolated from non-smoker participants in a previous study [30]. We observed an inflammatory response elicited by *S. aureus*, as measured by the secretion of the cytokines IL8, IL1β, IL6, and TNFα (Figure 6), for the 512 strain, which is not as abundant in the oral cavity as the resident 528-11 strain (21). It has been shown that inflammatory cytokines such as IFNγ and IL1β promote *S. aureus* colonization of the skin through biofilm formation and growth [45]. In addition, our data suggest that e-cig vape will dampen the initial immune response, allowing for *S. aureus* colonization (Figure 4 and Figure 5) of the oral cavity with a possible consequence of inducing oral dysbiosis.

The main virulence factors of *S. aureus* to cause tissue injury and inflammation include SpA, αtoxin, βtoxin, and PVL [46]. Mechanistically, SpA can activate the proinflammatory signaling cascade resulting in the activation of MAPK, and the transcriptional factor NFκB [46]. We demonstrate that *S. aureus* forms biofilm in response to e-cig aerosols (Figure 4). *S. aureus* attachment to OKF6 cells is also increased upon pre-exposure of *S. aureus* to e-cig aerosols (Figure 5), indicating that genes relating to colonization may have been induced. The increased attachment of *S. aureus* upon pre-exposure to e-cig is similar to the observations in previous studies regarding carriage in smokers compared with healthy non-smokers [44].

We also observed enhanced *S. aureus* growth in co-culture with oral epithelial cells pre-exposed to e-cig aerosols (Figure 5), indicating that e-cig aerosols impaired their ability to clear *S. aureus* “carriage” compared with untreated co-cultures, implying a potential immunosuppression. To further explore the effect of e-cig aerosols on the inflammatory responses of epithelial cells, we grew epithelial cells with e-cig aerosol and *S. aureus*, and measured the secretion of the cytokines IL8, IL1β, IL6, and TNFα (Figure 7). E-cig aerosol decreased the secretion of IL1β, IL8, and IL6, supporting the idea that e-cig aerosols potentially impair the host immune response against *S. aureus*. We speculate that the observed discrepancy in results between 512 and 528 is due to differences in bacterial–cell interactions resulting in less secretion of cytokines after co-culture with 528-11. Strain 528-11 is known to be a resident of the oral cavity and therefore may not induce as strong of an inflammatory response [21]. In addition, Cole et al. also determined previously that the SpA content of 528-11 was low compared with most nasal *S. aureus* strains [39].

*S. aureus* has also been shown to be enriched in oral squamous cell carcinoma [47,48] but its role in cancer development and progression remains unknown. To further explore the potential effects of *S. aureus* and e-cig aerosols in the context of cancer we explored the expression of COX2, which is an inflammatory marker known to play a key role in cancer development [41]. When oral epithelial cells were co-cultured with *S. aureus* and exposed to e-cig aerosols, we observed an increase in COX2 expression (Figure 6). Another recent study has shown that infection of HOK cells with *S. aureus* can facilitate increased cell proliferation through upregulation of COX2 and a concomitant increase in cyclin D1 with a suppression of p16. These changes resulted in colony formation growth as a measure of malignancy [49].

DNA damage is another important aspect in cancer development [50]. Exposure to DNA-damaging agents results in structural changes, the most dramatic of which are DNA double-strand breaks. If unrepaired, they can cause apoptosis. If misrepaired, they can lead to mutations and chromosomal translocation, which are the first steps in carcinogenesis. We established that aerosolized e-cig vape induced DNA damage (Figure 2), and we wanted to explore whether the presence of *S. aureus* with and without e-cig exposure would cause DNA damage in this experimental setting. We observed that *S. aureus* alone induced DNA damage compared with the untreated control (Figure 7). Consistent with this, we also observed an increase in DNA damage when oral epithelial cells were exposed to e-cig aerosols and co-cultured with *S. aureus* (Figure 7). Therefore, the increase in *S. aureus* colonization after exposure to e-cig aerosols could indirectly confer *S. aureus* an advantage to invade the tissue where it can further induce DNA damage in the host, potentially playing a role in cancer initiation. pH2AX foci mark sites of double-strand DNA breaks and subsequently recruit a complex of additional proteins involved in DNA damage response and repair. At the completion of proper DNA repair, pH2AX is typically de-activated [51]. pH2AX can remain activated if cells continue to replicate without proper DNA repair [52]. Our assessment of DNA damage focused on pH2AX (Figure 2 and Figure 7) as it has shown to be higher in dysplastic oral tissues and significantly associated with disease progression to oral cancer [53].

Oral cancer is the most common malignant epithelial neoplasm affecting the mouth and throat and is one of the leading sites for new cancer cases in 2020 (4% estimated new cases), according to the American Cancer Society. The main risk factors for tumors in the oral cavity are alcohol consumption and tobacco smoking, resulting in cancer with poor prognosis and survival [54] compared with tumors of the oropharynx, which are mostly caused by human papilloma virus (HPV) infection [55], and have better outcome and survival than HPV-negative lesions of the oral cavity. Although cigarette smoking prevalence has decreased over the past decades, the “Population Assessment of Tobacco and Health” study, a nationally representative, longitudinal cohort study of US youth and adults, regarding the first three waves (2013–2016), shows that e-cig and poly-tobacco use (use of more than one tobacco product) are increasing dramatically [1]. While initially marketed as an aid in smoking cessation and a cleaner alternative to tobacco smoking [3], e-cigs have attracted middle school and high school students and their use is associated with continued cigarette smoking behavior and nicotine dependence among youth and adults [1]. Furthermore, the “e-cig epidemic” has raised serious concerns about the health risks and carcinogenic potential of e-cig aerosols. Given the current numbers for e-cig users (about 3 million teens and 10 million adults), the effect of e-cig aerosols on oral epithelial cells and host–*S. aureus* interactions in the promotion of chronic inflammation and associated carcinogenesis is an important research area.

## 5. Conclusions

E-cig vapor, regardless of nicotine content, induced DNA damage in oral epithelial cells, along with a decrease in proliferation and cell viability. While the initial immune response is dampened in the presence of e-cig aerosols, under chronic conditions, e-cig aerosols have the capacity to induce an inflammatory response and COX2-mediated signaling, which could lead to cancer initiation. Therefore, we conclude that the usage of e-cigs can affect the oral epithelium by directly inducing an inflammatory response and DNA damage. However, even more interestingly, we find that e-cig aerosols affect oral bacteria, such as *S. aureus*, enhancing their colonization capacity and pathogenicity, inducing DNA damage as a consequence.

## Figures and Tables

**Figure 1 cells-11-00773-f001:**
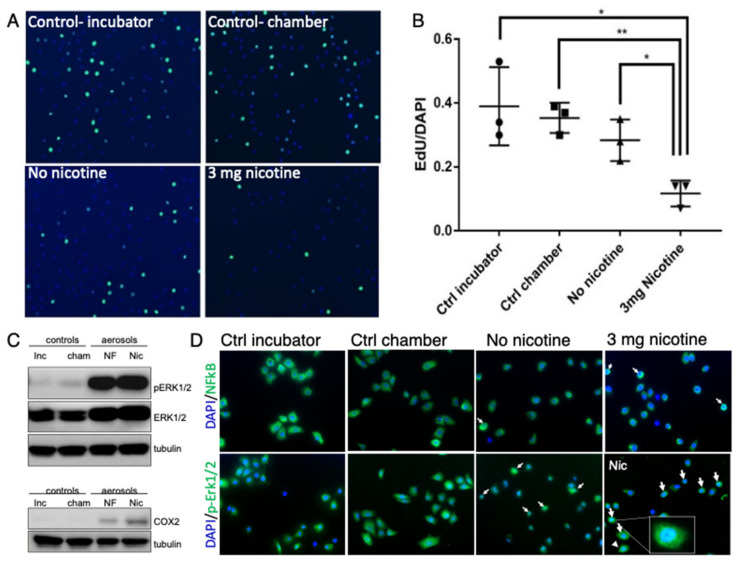
E-cig vape reduces cell proliferation and activates proinflammatory pERK1/2 and NF-kB signaling. Epithelial oral immortalized OKF6 cells were exposed to nicotine-free (NF) and nicotine-containing (nicotine/Nic, 3 mg/mL) e-cig aerosol, and compared to air in the vape chamber (chamber) and incubator controls. Click-iT EdU staining was utilized to detect (**A**) and quantify (**B**) changes in proliferation. Phospho-specific ERK1/2 antibody was used to detect signaling activation compared to total ERK1/2 and COX2 expression in protein lysates from STR cells (**C**). STR cells were fixed 3 h after vape exposure and used for immunofluorescence staining with antibody against pERK1/2 and NF-kB (**D**) to show nuclear translocation of pERK1/2 and NF-kB. White arrows point to nuclear detection of pERK1/2 and NF-kB. * *p* < 0.05, ** *p* < 0.01.

**Figure 2 cells-11-00773-f002:**
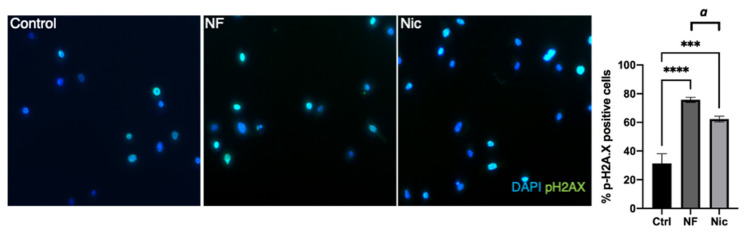
E-cig aerosol induces DNA damage. OKF6 cells were exposed to nicotine-free (NF) and nicotine-containing (Nic 3 mg/mL) e-cig aerosol compared with control. Phosphorylation of the histone protein H2AX was measured by immunofluorescence. Representative images are shown. Data were collected by counting positive cells (pH2AX) in images and building a ratio over total cells per image (DAPI). *** *p* < 0.001, **** *p* < 0.0001 (ANOVA), ***a*** *p* < 0.001 (Student’s *t*-test).

**Figure 3 cells-11-00773-f003:**
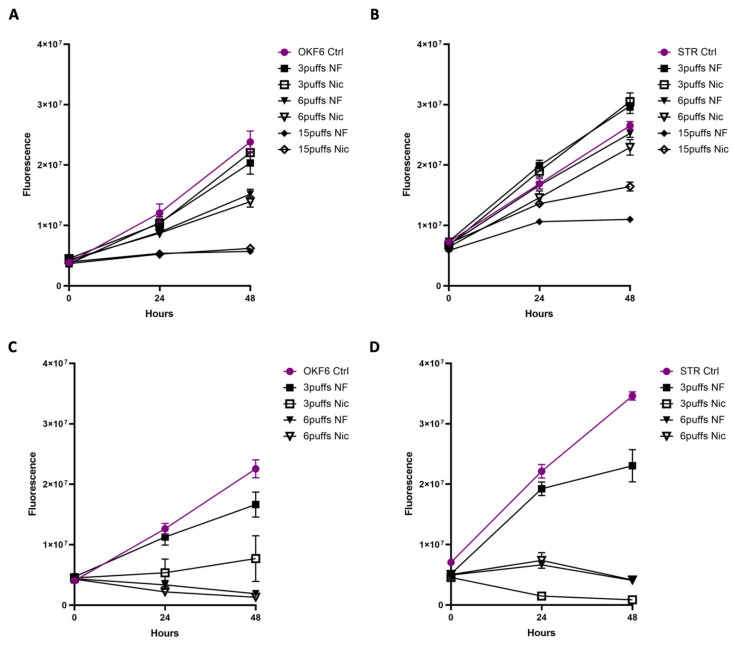
Exposure to e-cig aerosols decreases epithelial cell viability in a dose-dependent manner. Acute exposure of OKF6 (**A**) and STR (**B**) cells to e-cig aerosol pretreated media (6 puffs, 3 s length, 30 s intervals) for 30 min followed by a recovery of 48 h in fresh media. Cell viability was measured every 24 h with cell titer blue. Chronic exposure of OKF6 (**C**) and STR (**D**) to e-cig aerosol pretreated media for 48 h (6 puffs, puff length of 3 s, intervals between puffs of 30 s). Cell viability was measured every 24 h with cell titer blue.

**Figure 4 cells-11-00773-f004:**
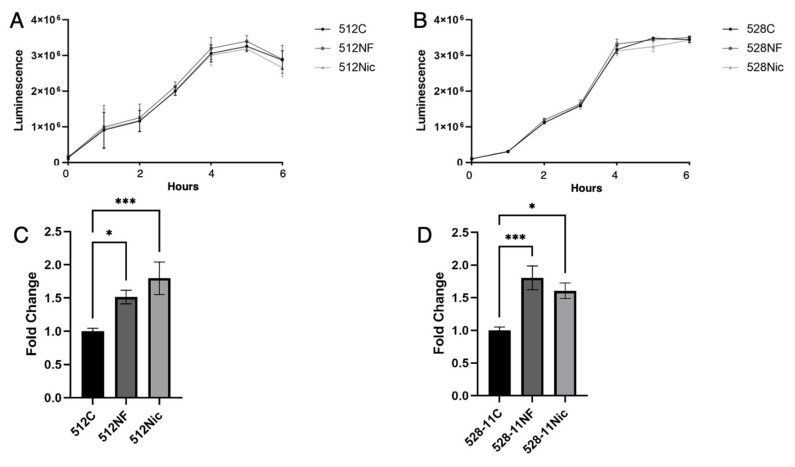
*S. aureus* growth was unaffected by e-cig aerosol exposure, but biofilm formation was enhanced. *S. aureus* strains 512 (**A**) and 528-11 (**B**) were chronically (continuously) exposed to e-cig aerosol pretreated media for 24 h (6 puffs, puff length of 3 s, intervals between puffs of 30 s). Cell viability was measured every hour with BacTiter Glo (Promega). Biofilm formation of *S. aureus* 512 (**C**) and 528-11 (**D**) was measured after 24 h incubation in e-cig aerosol pretreated media. * *p* < 0.05, *** *p* < 0.001.

**Figure 5 cells-11-00773-f005:**
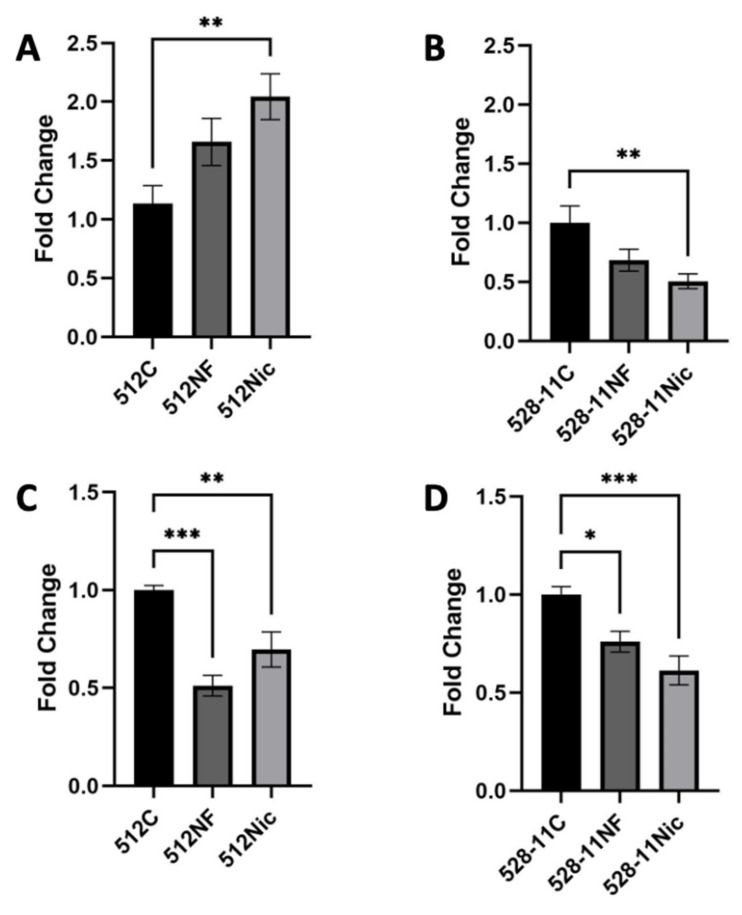
E-cig aerosol exposure enhances *S. aureus* colonization. *S. aureus* attachment is increased after e-cig aerosol exposure. OKF6 cells were co-cultured with e-cig aerosol pretreated *S. aureus* (MOI 1, 3 h co-culture) and attachment of *S. aureus* 512 (**A**) and 528-11 (**B**) was measured. C—control; NF—nicotine-free; Nic—3 mg/mL nicotine. E-cig aerosol exposure reduces *S. aureus* clearance. OKF6 cells were exposed to e-cig aerosol and co-cultured with e-cig aerosol pretreated *S. aureus* (MOI 1, 3 h co-culture) and clearing of *S. aureus* 512 (**C**) and 528-11 (**D**) was measured. * *p* < 0.05, ** *p* < 0.01, *** *p* < 0.001.

**Figure 6 cells-11-00773-f006:**
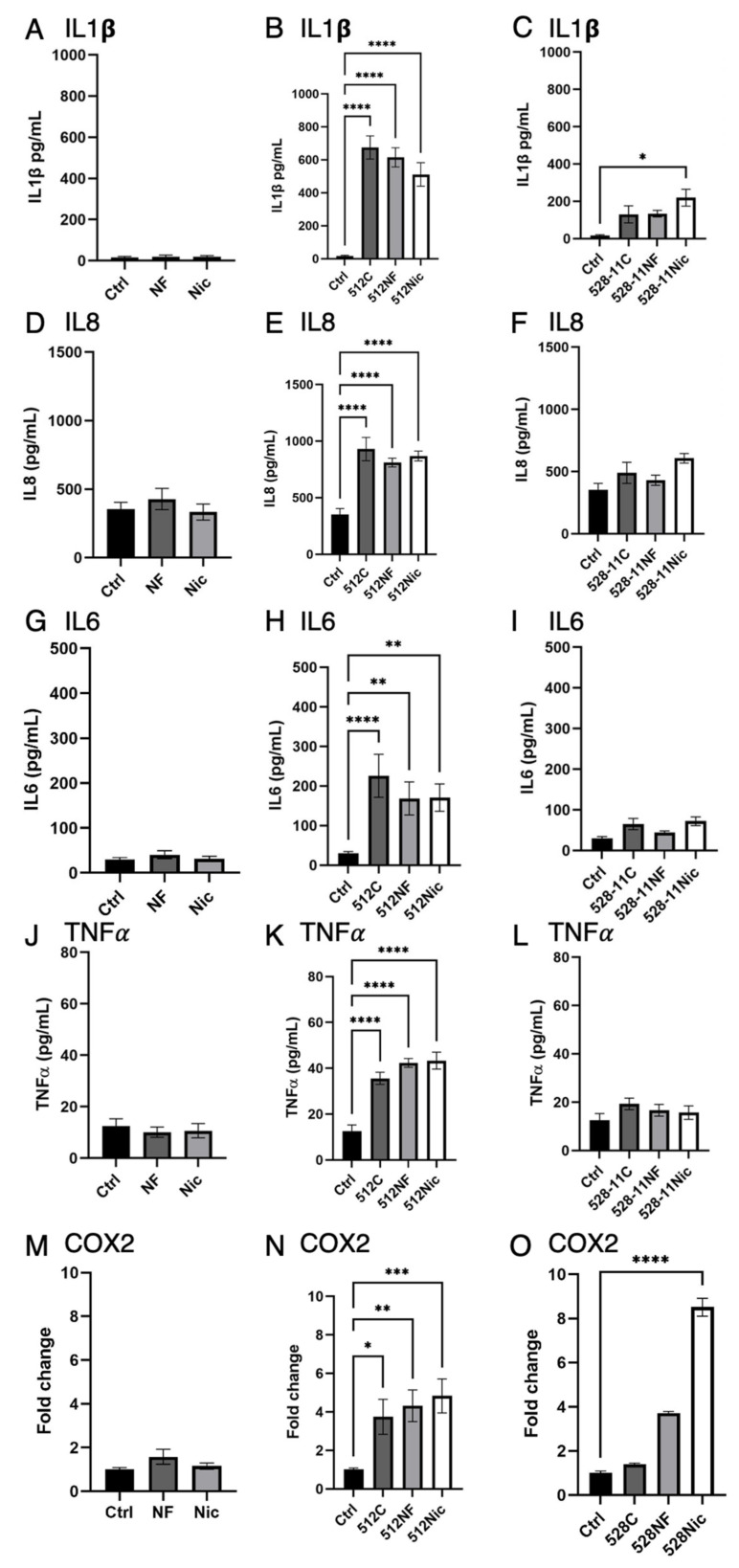
Secretion of the proinflammatory cytokines IL1β (**A**–**C**), IL8 (**D**–**F**), IL6 (**G**–**I**), and TNFα (**J**–**L**) was measured by ELISA: e-cig aerosols decreased the release of proinflammatory cytokines IL1β, IL8, and IL6 when OKF6 cells were pre-exposed to e-cig aerosols 30 min (6 puffs 30 s interval; C—control; NF—nicotine-free; Nic—3 mg/mL nicotine) and co-cultured with *S. aureus 512* (**B**,**E**,**H**,**K**) and *S.aureus 528-11* (**C**,**F**,**I**,**L**) at MOI 10^−2^. Expression of COX2 was measured by qRT-PCR after e-cig aerosol exposure (**M**) and co-culture with *S. aureus* (**N**,**O**). * *p* < 0.05, ** *p* < 0.01, *** *p* < 0.001, **** *p* < 0.0001.

**Figure 7 cells-11-00773-f007:**
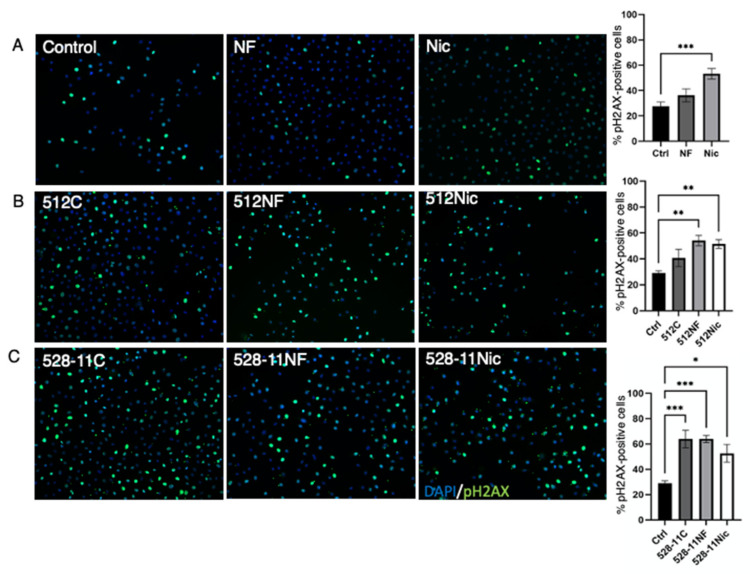
E-cig aerosol and *S. aureus* colonization of OKF6 cells induced DNA damage. OKF6 cells were pretreated with e-cig aerosol (**A**) and co-cultured with e-cig aerosol pretreated *S. aureus* strains 512 (**B**) and 528-11 (**C**), (MOI 1), 3 h co-culture; C—control; NF—nicotine-free; Nic—3 mg/mL nicotine. Phosphorylation of the histone protein H2AX was measured by immunofluorescence. Representative images are shown. Data were collected by counting positive cells (pH2AX, green) in images and building a ratio over total cells per image (DAPI, blue). * *p* < 0.05, ** *p* < 0.01, *** *p* < 0.001.

**Table 1 cells-11-00773-t001:** Primer sequences used for gene amplification in qRT-PCR.

Gene	5′ to 3′ Forward	5′ to 3′ Reverse
IL1β	GGA GAT TCG TAG CTG GAT GC	GAG CTC GCC AGT GAA ATG AT
COX2	TGA GCA TCT ACG GTT TGC TG	TGC TTG TCT GGA ACA ACT GC
IL8	CCT GAT TTC TGC AGC TCT GTG	CCA GAC AGA GCT CTC TTC CAT
TNFα	ACA AGC CTG TAG CCC ATG TT	AAA GTA GAC CTG CCC AGA CT

## Data Availability

No reporting of large data set.

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
