# Peer review of "E-Cigarette Aerosols Promote Oral S. aureus Colonization by Delaying an Immune Response and Bacterial Clearing"

_cells, 2022, doi:10.3390/cells11050773_

Round 1
Reviewer 1 Report
The manuscript describes an interesting and comprehensive study of the effect of e-cigarette vapor on oral epithelial cells alone and in combination with Staphylococcus aureus cells. As an in-vitro study it has merit. It would also be interesting to further analyse not only S. aureus, but a polymicrobial culture in vitro or test the influence of e-cigarette vapor in situ, resp. in vivo, to incorporate the effects and interactions of the polymicrobial nature of the oral microbiota.
Minor changes are advised:
L71-78 should be moved to the beginning of the discussion section
L125: the calculation/ formula is not clear
L 292-L299: this is rather content for the introduction section
L 328-332: this also belongs in the introduction section
L 352-356: this also belongs in the introduction section
L 394-401: this also rather belongs in the introduction section
Fig 6: it would be more reader-friendly to include the cytokine-name in each figure of the panel, not only in the legend
Fig. S2: also here, it would be more reader-friendly to include the cytokine name in each small figure.
Author Response
Dear Reviewer 1, Thank you for the overall positive feedback. We addressed the comments in a point-by-point manner:
L71-78 should be moved to the beginning of the discussion section
Response: Thank you for the feedback. We moved this section to new lines 435-443 at the beginning of the discussion.
L125: the calculation/ formula is not clear
Response: We deleted the formular and the related text as calculating MOI is standard molecular practice and we may have provided more complicated explainations than needed to understand the determination of the MOI. In short, it is based on the equatin (C1)(V1)=(C2)(V2)
The volume of bacteria (V1) used to inoculate a selected final volume (V2) was calculated using the equation below. Bacteria was taken from a snap with a known concentration (C1). The concentration of bacteria in the co-culture was determined based on the concentration of oral cells seeded and the MOI selected (C2).
L 292-L299: this is rather content for the introduction section
Response: Text about e-cigarette aerosols and their effect on the microbiota has been moved to new lines 64-72 in the introduction.
L 328-332: this also belongs in the introduction section
Response: We deleted the paragraph about biofilm formation in this context as we agree that this information is not necessary for the reader to follow the experimental design yet it also did not seem to add to the understand the rationale of the study.
L 352-356: this also belongs in the introduction section
Response: The inflammatory cytokine signaling background indeed fits better in conjunction with the new background on host responses S. aureus. New lines 72-76
L 394-401: this also rather belongs in the introduction section
Response: We agree with the reviewer, however, the first two sentences provide an introduction to why we assessed DNA damage in this context, so these remained unchanged and lines 397-402 where moved to the introduction: new lines 77-82
Fig 6: it would be more reader-friendly to include the cytokine-name in each figure of the panel, not only in the legend
Response: Added, thank you.
Fig. S2: also here, it would be more reader-friendly to include the cytokine name in each small figure.
Response: Corrected, thank you.
Reviewer 2 Report
The manuscript by Cátala-Valentín et al focus on the study of the effect of e-cig aerosols on Staph aureus growth onto two epithelial cell lines (OKF6- oral, and STR- esophageal. Although the manuscript is scientifically sound and interesting, its readability should be improved. It is sometimes not clear whether, what or why they have used what they used in some experiments.
Specific comments to take into account are the following:
- Line 89: Bacterial cultures- Why have the authors focused on studying 512 and 528-11 Staph aureus strains in the rpesent study? Is it related to their attachment capacity shown in Figure S1?
- Line 105- Electronic cigarette aerosol- How this incubation time was determined? Have you performed time-course experiments? What is the average time that oral cells re exposed to e-vape?
- Line 108- e-cigarette liquid was obtained from 2 companies (Vaporfi ad Vapor Vapes). Which one and for what is used in this study? Have you used both and compared them? It is currenty not clear.
- Line 113- What do you mean with topography?
- Line 135: Cell Titer Blue- Is this linear relationship between cell number and fluorescence specified by the manufacturer? Have you determined this?
- Line 175: Biofilm formation- Why is the removal of planktonic bacteria performed?
- Line 184: ELISA- hBD-3, plese, write in full.
- Line 232: Statistical analyses- I think the authors maen "Dunn's (not Dunnet's) correction". Which software was used for statistical analyses? Please, write it here.
- Lines 238 and 241- Are vape chamber and smoke chamber the same? Please, clarify.
- Figure 2. Are there any significant differences between NF and Nic? It seems the case according to the graph. Please, revise and correct accordingly.
- Figure 4C and D. How was biofilm formation measured? Why is fold change depicted in the graphs? What about absorbance?
- Figure 5B. Any comments on why the effect of e-cig aerosol exposure on the two Staph aureus strans in study is different? The authors claim that " S. aureus attachment is increased after e-cig aerosol exposure", but the pattern is the opposite for the 2 strains. Could you please clarify this?
- Figure 6C, F and I. Can you please speculate on whay do you think the pattern is different to the other S. aureus strain?
- Line 409. Figure 7C and D. Please, remove D. There is no Figure 7D.
- Figure 7B and C. Again the pattern between the two strains is different here. Why could that be, in your opinion?
Author Response
Dear Reviewer 2,
We would like to thank the reviewer for the comprehensive feedback and the in-depth guidance on how to improve the manuscript. We addressed each comment in a point-by-point manner.
Line 89: Bacterial cultures- Why have the authors focused on studying 512 and 528-11 Staph aureus strains in the present study?
Response: Passarielo C et al (ref 21) reported that the sequence types ST30 (512) and ST8/t008 (528-11) strains are prevalent isolates from the oral cavity with ST8/t008 less abundant than ST30 (see more detail below) and associated with active sites of periodontitis- added detail to section 3.3 line 317-318. We therefore chose to assess these two bacteria as models for S. aureus colonization yet each strain is known to be different, e.g. SpA status, and the host repsonses therefore not the expected to be the same (see below). These differences were also pointed out in the discussion (lines 389-to 394).
Line 105- Electronic cigarette aerosol- How this incubation time was determined? Have you performed time-course experiments? What is the average time that oral cells re exposed to e-vape?
Response: As stated in the Methods section (line 132-137), the e-cig exposure times were determined based on the vaping habits (puff topography) of established cigarette smokers (Y.O. Lee, et al 2018). We performed a cell titer to evaluate how different amount of puffs affected the viability of oral cells (Figure 3), and based on those experiments we selected 6 puffs for subsequent experiments as the epithelial cells showed an initial growth delay but recovered.
Line 108- e-cigarette liquid was obtained from 2 companies (Vaporfi and Vapor Vapes). Which one and for what is used in this study? Have you used both and compared them? It is currently not clear.
Response: Figure numbers were added to the materials and methods sections 2.3 electronic cigarette aerosols to identify the experiments performed with e-cigarette liquids from each vendor. Only experiments show in Figure 1(122) used Vaporfi e-cigarette liquid (line 12126). For all subsequent experiments (Figures 2-7, line 126) Vapor Vapes was used since Vapor-Fi was no longer available for purchase.
Line 113- What do you mean with topography?
Response: An e-cigarette users’ puffing behaviors is often referred to as topography and is reflected in parameters such as the puff volumes and puff durations. Line 53.
Line 135: Cell Titer Blue- Is this linear relationship between cell number and fluorescence specified by the manufacturer? Have you determined this?
Response: This relationship is listed by the manufacturer and was not tested by us. We updated the text to refer to this source (now line 151-152).
Line 175: Biofilm formation- Why is the removal of planktonic bacteria performed?
Response: The planktonic bacteria are removed as they remain free floating and did not participate in biofilm formation. Safranin staining would still provide a positive ‘signal’ for these bacteria and contribute to the overall measurement, which would skew the assessment and enumeration of biofilm forming bacteria.
Line 184: ELISA- hBD-3, please, write in full.
Response: Corrected to human beta defensin-3, now line 200.
Line 232: Statistical analyses- I think the authors mean "Dunn's (not Dunnet's) correction". Which software was used for statistical analyses? Please, write it here.
Response: The software utilized to perform the statistical analyses was GraphPad Prism 9. For the statistical analysis we performed using one-way ANOVA with a Dunnett’s correction (multiple comparsion test), which is generally used when testing two or more experimental groups against the same control group, as we did in all of the experiments. (Lee, S., & Lee, D. K. (2018). What is the proper way to apply the multiple comparison test?. Korean journal of anesthesiology, 71(5), 353–360. https://doi.org/10.4097/kja.d.18.00242)
Lines 238 and 241- Are vape chamber and smoke chamber the same? Please, clarify.
Response: We apologize for the confusion. Yes, vape and smoke chamber are the same and it is now consistently called vape chamber throughout the manuscript.
Figure 2. Are there any significant differences between NF and Nic? It seems the case according to the graph. Please, revise and correct accordingly.
Response: For the statistical analysis we performed a one-way ANOVA with Dunnett’s correction to compare different experimental groups against the same control group, in this case against the untreated control. Based on this analysis, the significance between NF and Nic cannot be determined. When an individual t-test was performed between NF and Nic groups, a statistically significant difference was observed. We didn’t want to mix two different statistical approaches in the same figure and give the interest of the effect of vape in general focused on the ANOVA comparing both to control. Please, let us know if you would like us to add the test to the Figure.
Figure 4C and D. How was biofilm formation measured? Why is fold change depicted in the graphs? What about absorbance?
Response: As described in the Materials and Methods (section 2.7, line 190 and below), the biofilm forms over 24 hours and is stained with safranin. Once dissociated the in ethanol/acetone, absorbance of the safranin solution is measured at OD490. We set the OD for the control to 1 to allow for a direct comparison of the effect of treatments.
Figure 5B. Any comments on why the effect of e-cig aerosol exposure on the two Staph aureus strains in study is different? The authors claim that " S. aureus attachment is increased after e-cig aerosol exposure", but the pattern is the opposite for the 2 strains. Could you please clarify this?
Response: Thank you for pointing to the differential response. We did alter the text to account for the different outcome in attachment. Line 342
As stated below also for questions relating to Figures 6 and 7: In lines 389-to 394 we speculate on the different reasons for the observed discrepancy in results between 512 and 528: 528 is known to be a more abundant resident of the oral cavity and therefore may not induce as strong of an inflammatory response (ref: C. Passariello, A. Lucchese, A. Virga, F. Pera, and P. Gigola, “Isolation of Staphylococcus aureus and progression of periodontal lesions in aggressive periodontitis,” Eur. J. Inflamm., vol. 10, no. 3, pp. 501–513, 2012 ). In addition, Cole et al. also determined previously that the SpA content of 528-11 was low compared to most nasal S. aureus strains most likely the cause for a suppressed host response.
Figure 6C, F and I. Can you please speculate on why do you think the pattern is different to the other S. aureus strain?
Figure 7B and C. Again the pattern between the two strains is different here. Why could that be, in your opinion?
Response: In lines 389-to 394 we speculate on the different reasons for the observed discrepancy in results between 512 and 528: 528 is known to be a more abundant resident of the oral cavity and therefore may not induce as strong of an inflammatory response (ref: C. Passariello, A. Lucchese, A. Virga, F. Pera, and P. Gigola, “Isolation of Staphylococcus aureus and progression of periodontal lesions in aggressive periodontitis,” Eur. J. Inflamm., vol. 10, no. 3, pp. 501–513, 2012 ). In addition, Cole et al. also determined previously that the SpA content of 528-11 was low compared to most nasal S. aureus strains most likely the cause for a suppressed host response.
Line 409. Figure 7C and D. Please, remove D. There is no Figure 7D.
Response: Corrected. Thank you!
Round 2
Reviewer 2 Report
The authors have replied to all my raised comments. However, there are still some points to correct:
Introduction: Lines 46 and 64 are the same. Please remove one or rephrase accordingly.
M&M: Line 107- Please, include a short explanation why the study focuses on 512 and 528-11 S aureus strains. It is still missing. I also do not see that the differences in response by these 2 strains are included in the discussion in the current revised version. Please, include.
Passarielo C et al (ref 21) reported that the sequence types ST30 (512) and ST8/t008 (528-11) strains are prevalent isolates from the oral cavity with ST8/t008 less abundant than ST30 (see more detail below) and associated with active sites of periodontitis- added detail to section 3.3 line 317-318. We therefore chose to assess these two bacteria as models for S. aureus colonization yet each strain is known to be different, e.g. SpA status, and the host repsonses therefore not the expected to be the same (see below). These differences were also pointed out in the discussion (lines 389-to 394).
Figure 2- Regarding the significant differences between NF and Nic, I would still include the significance in the graph, and the test used for comparison between these two in the statistical methods and the Figure 2 legend.
Figure 4C and D- Please, include this information in the corresponding M&M section: We set the OD for the control to 1 to allow for a direct comparison of the effect of treatments.
Author Response
Thank you again for your detailed guidance in how to improve the manuscript. We modified the text and figure 2 in response to this reviewer’s comments and suggestions and highlighted the latest changes in yellow throughout the manuscript.
Introduction: Lines 46 and 64 are the same. – Response: Thank you for pointing that out!
M&M: Line 107- Please, include a short explanation why the study focuses on 512 and 528-11 S aureus strains. It is still missing. I also do not see that the differences in response by these 2 strains are included in the discussion in the current revised version. Please, include.
Response: New text is included (lines 109-111, 321-324, 396-401, discussion 483-488).
Figure 2- Regarding the significant differences between NF and Nic, I would still include the significance in the graph, and the test used for comparison between these two in the statistical methods and the Figure 2 legend.
Response: Figure 2 and the legend is modified to include the additional statistical analysis. Materials and Methods were also edited to refer to t-test as an additional statistical method used. Lines 251 and line 289.
Figure 4C and D- Please, include this information in the corresponding M&M section: We set the OD for the control to 1 to allow for a direct comparison of the effect of treatments.
Response: Added (line 199), thank you.
Thank you for your consideration.